# Softmax is $1/2$-Lipschitz: A tight bound across all $\ell_p$ norms

**Pravin Nair**                                                                                   *pravinnair@ee.iitm.ac.in*
*Department of Electrical Engineering*
*Indian Institute of Technology, Madras*

**Reviewed on OpenReview:** *https://openreview.net/forum?id=6dowaHsa6D*

## Abstract

The softmax function is a basic operator in machine learning and optimization, used in classification, attention mechanisms, reinforcement learning, game theory, and problems involving log-sum-exp terms. Existing robustness guarantees of learning models and convergence analysis of optimization algorithms typically consider the softmax operator to have a Lipschitz constant of 1 with respect to the $\ell_2$ norm. In this work, we prove that the softmax function is contractive with the Lipschitz constant $1/2$, uniformly across all $\ell_p$ norms with $p \geq 1$. We also show that the local Lipschitz constant of softmax attains $1/2$ for $p = 1$ and $p = \infty$, and for $p \in (1, \infty)$, the constant remains strictly below $1/2$ and the supremum $1/2$ is achieved only in the limit. To our knowledge, this is the first comprehensive norm-uniform analysis of softmax Lipschitz continuity. We demonstrate how the sharper constant directly improves a range of existing theoretical results on robustness and convergence. We further validate the sharpness of the 1/2 Lipschitz constant of the softmax operator through empirical studies on attention-based architectures (ViT, GPT-2, Qwen3-8B) and on stochastic policies in reinforcement learning.

## 1 Introduction

The softmax function has applications across diverse areas of machine learning and optimization. In classification models, model output logits are normalized using the softmax function to form a probability distribution (Goodfellow et al., 2016). In Transformer architectures, softmax normalizes attention scores to compute weighted combinations of input token features, enabling high-quality feature refinement (Vaswani et al., 2017). These transformer architectures have shown state-of-the-art performance in various applications in natural language processing (Tunstall et al., 2022), computer vision (Khan et al., 2022), reinforcement learning (Li et al., 2023), etc. Since softmax is the gradient of the log-sum-exp function, its properties are crucial for analyzing optimization algorithms (Gao & Pavel, 2017; Nachum et al., 2017; Nesterov, 2005). In reinforcement learning, softmax is used to convert action-value estimates (Q-values) to stochastic policies. This results in a probability distribution over actions that favours actions with higher Q-values and still assigns probabilities to suboptimal actions. In entropy-regularized reinforcement learning, the softmax operator arises naturally as the solution of the policy optimization problem (Sutton & Barto, 2018; Nachum et al., 2017).

The softmax operator maps any real vector to a probability distribution. Formally, the softmax function $\sigma_\lambda : \mathbb{R}^n \to \Delta_n^\circ$ with inverse–temperature parameter $\lambda > 0$ is defined as

$$\sigma_\lambda(\boldsymbol{x})_i \;=\; \frac{\exp(\lambda x_i)}{\sum_{j=1}^n \exp(\lambda x_j)}, \qquad i = 1, \dots, n, \tag{1}$$

where $\Delta_n = \{\boldsymbol{u} \in \mathbb{R}^n : u_i \geq 0, \sum_i u_i = 1\}$ is the probability simplex and $\Delta_n^\circ = \{\boldsymbol{u} \in \mathbb{R}^n : u_i > 0, \sum_i u_i = 1\}$ denotes its interior. Since $\exp(x_i) > 0$ for all $x_i \in \mathbb{R}$, $\sigma_\lambda(\boldsymbol{x})$ never attains the boundary of the simplex $\partial \Delta_n = \{\boldsymbol{u} \in \mathbb{R}^n : u_i = 0 \text{ for some } i\}$. The coefficient $\lambda$ measures the smoothness of the output distribution. A larger $\lambda$ makes the distribution more peaked on the largest entries, and a smaller $\lambda$ makes it smoother with more evenly spread probability values. The standard softmax function is when $\lambda = 1$.

Accurate analysis of softmax's Lipschitz constant has important applications across machine learning and optimization. It enables accurate robustness analysis of attention mechanisms, where Lipschitz bounds can be used to set hyperparameters that ensure stable training (Qi et al., 2023). In game-theoretic learning, modeling action selection via softmax allows Lipschitz continuity to be directly leveraged for establishing convergence rates toward Nash equilibria (Gao & Pavel, 2017). Many existing generalization results for learning models require computing the Lipschitz constant of the loss function with respect to either network input or network parameters. When softmax appears in the final layer or in attention, an accurate Lipschitz analysis becomes essential for deriving sharp generalization bounds (Asadi & Abbe, 2020). Moreover, when a model is expressed as a composition of layers, standard network analyses bound the global Lipschitz constant by an appropriate composition (often a product) of layer-wise constants Bartlett et al. (2017); Miyato et al. (2018); Virmaux & Scaman (2018); Fazlyab et al. (2019). In such bounds, the softmax Lipschitz constant acts as a multiplicative factor, so tightening it leads directly to sharper network-level guarantees. In classification networks, robustness and domain adaptation depend on how sensitive output probabilities are to input perturbations, which is again controlled by the Lipschitz constant of softmax (Chen et al., 2022). The log-sum-exp (LSE) function is a smooth approximation of the maximum operator, and the gradient of LSE is the softmax function. Hence, in optimization problems with LSE-based objective functions, step-size rules for gradient descent and smoothness-based convergence rates can be sharpened using a tight softmax Lipschitz bound. Lipschitz constant of softmax is also used in analysing several other frameworks, including meta-learning (Jeon et al., 2024), structural causal models (Le Priol et al., 2021), prior-data fitted networks (Nagler, 2023), and sparse training (Lei et al., 2024). The above-mentioned diverse set of applications highlights that deriving a tight Lipschitz constant of softmax has a substantial practical impact.

Despite the theoretical and empirical significance of softmax's Lipschitz constant, an accurate analysis is missing in the literature. A number of works (Gao & Pavel, 2017; Laha et al., 2018; Asadi & Abbe, 2020; Gouk et al., 2021; Le Priol et al., 2021; Chen et al., 2022; Nagler, 2023; Jeon et al., 2024; Lei et al., 2024) assume the softmax function to be 1-Lipschitz with respect to the $\ell_2$ norm. This assumption is often attributed to Proposition 4 in Gao & Pavel (2017). In some works, this is mentioned as a straightforward fact, perhaps because showing the Lipschitz constant to be bounded by 1 is trivial. We also note that Xu et al. (2022) reports a bound of $1/4$ for the softmax's Lipschitz constant. The main motivation for this work is the discrepancy between the commonly assumed Lipschitz constant of the softmax function and the smaller values observed in our empirical studies on transformer architectures. This prompted us to derive a tighter bound for the Lipschitz constant of the softmax function. In this regard, our contributions are as follows:

- We prove that softmax is $1/2$-Lipschitz across all $\ell_p$ norms ($p \geq 1$), thereby improving the usually assumed bound of 1.

- Furthermore, we show the tightness of this bound by proving that the local Lipschitz constant is attained for $p = 1$ and $p = \infty$ as $1/2$, while for $p \in (1, \infty)$ the local Lipschitz constants remain strictly below $1/2$ and the supremum is only approached in the limit.

- We demonstrate that our tight Lipschitz bound improves the existing theoretical results in robustness and convergence. We also validate our derived Lipschitz constant through experiments on large-scale transformer models and reinforcement learning policies.

## 2 Preliminaries

In this section, we provide the necessary background for our main results. We start with the definition of the $\ell_p$ norm for vectors and the corresponding induced operator norm for matrices.

*Definition* 2.1 ($\ell_p$ norm). For a vector $\boldsymbol{x} = (x_1, x_2, \ldots, x_n) \in \mathbb{R}^n$ and for $1 \leq p < \infty$, the $\ell_p$ norm of $\boldsymbol{x}$ is defined as

$$\|\boldsymbol{x}\|_p = \left( \sum_{i=1}^{n} |x_i|^p \right)^{1/p}.$$

For $p = \infty$, the $\ell_\infty$ norm is defined as

$$\|\boldsymbol{x}\|_\infty = \max_{1 \leq i \leq n} |x_i|.$$

We can also define the operator norm of a matrix $\boldsymbol{A} = (A_{ij}) \in \mathbb{R}^{m \times n}$ induced by the $\ell_p$ norm as

$$\|\boldsymbol{A}\|_p := \sup_{\boldsymbol{v} \neq 0} \frac{\|\boldsymbol{A}\boldsymbol{v}\|_p}{\|\boldsymbol{v}\|_p} = \sup_{\|\boldsymbol{v}\|_p = 1} \|\boldsymbol{A}\boldsymbol{v}\|_p.$$

Throughout the paper, we overload $\|\cdot\|_p$ to represent both the vector norm and the corresponding induced norm on matrices. Next, we define the Lipschitz property of functions.

*Definition* 2.2 (Lipschitz continuity in $\ell_p$ norm). Let $f : \mathbb{R}^n \to \mathbb{R}^m$ and let $\|\cdot\|_p$ denote the $\ell_p$ norm with $1 \leq p \leq \infty$. The function $f$ is said to be Lipschitz continuous with respect to $\|\cdot\|_p$ if there exists a constant $L_p \geq 0$ such that

$$\|f(\boldsymbol{x}) - f(\boldsymbol{y})\|_p \leq L_p \|\boldsymbol{x} - \boldsymbol{y}\|_p, \qquad \forall \boldsymbol{x}, \boldsymbol{y} \in \mathbb{R}^n. \tag{2}$$

The smallest such constant $L_p$ is called the Lipschitz constant of $f$ with respect to the $\ell_p$ norm.

A mapping $f$ is contractive if $L_p < 1$, and non-expansive if $L_p \leq 1$. It is firmly non-expansive if $\|f(\boldsymbol{x}) - f(\boldsymbol{y})\|_p^2 \leq \langle f(\boldsymbol{x}) - f(\boldsymbol{y}), \boldsymbol{x} - \boldsymbol{y} \rangle$ for all $\boldsymbol{x}, \boldsymbol{y}$, and co-coercive with constant $\beta > 0$ if $\langle f(\boldsymbol{x}) - f(\boldsymbol{y}), \boldsymbol{x} - \boldsymbol{y} \rangle \geq \beta \|f(\boldsymbol{x}) - f(\boldsymbol{y})\|_p^2$.

Note that the Lipschitz constant in Eq. 2 is a global quantity, since the inequality holds for all $\boldsymbol{x}, \boldsymbol{y} \in \mathbb{R}^n$. By definition, $L_p$ can be characterized as,

$$L_p = \sup_{\boldsymbol{x}, \boldsymbol{y} \in \mathbb{R}^n, \boldsymbol{x} \neq \boldsymbol{y}} \frac{\|f(\boldsymbol{x}) - f(\boldsymbol{y})\|_p}{\|\boldsymbol{x} - \boldsymbol{y}\|_p}. \tag{3}$$

We next introduce the notion of a local Lipschitz constant (Rockafellar & Wets, 1998) via the Jacobian of $f$, which in turn provides another characterization of the global Lipschitz constant.

*Definition* 2.3 (Jacobian matrix (Boyd & Vandenberghe, 2004)). Let $f : \mathbb{R}^n \to \mathbb{R}^m$ be a differentiable function, where $f(\boldsymbol{x}) = \big(f_1(\boldsymbol{x}), f_2(\boldsymbol{x}), \ldots, f_m(\boldsymbol{x})\big)$ and each $f_i : \mathbb{R}^n \to \mathbb{R}$ is scalar-valued function. The Jacobian matrix of $f$ at $\boldsymbol{x}$, denoted $\boldsymbol{J}_f(\boldsymbol{x}) \in \mathbb{R}^{m \times n}$, is the matrix of all partial derivatives,

$$\boldsymbol{J}_f(\boldsymbol{x}) = \begin{bmatrix} \frac{\partial f_1}{\partial x_1}(\boldsymbol{x}) & \cdots & \frac{\partial f_1}{\partial x_n}(\boldsymbol{x}) \\ \vdots & \ddots & \vdots \\ \frac{\partial f_m}{\partial x_1}(\boldsymbol{x}) & \cdots & \frac{\partial f_m}{\partial x_n}(\boldsymbol{x}) \end{bmatrix}.$$

If the function is continuously differentiable, the local Lipschitz constant of a function at any $\boldsymbol{x} \in \mathbb{R}^n$ is equal to the $\ell_p$ norm of the Jacobian at that point.

*Definition* 2.4 (Local Lipschitz constant (Rockafellar & Wets, 1998)). Let $f : \mathbb{R}^n \to \mathbb{R}^m$. The local Lipschitz constant of $f$ at a point $\boldsymbol{x} \in \mathbb{R}^n$ with respect to the $\ell_p$ norm is defined as

$$L_p(\boldsymbol{x}) := \limsup_{\boldsymbol{y} \to \boldsymbol{x}, \, \boldsymbol{y} \neq \boldsymbol{x}} \frac{\|f(\boldsymbol{y}) - f(\boldsymbol{x})\|_p}{\|\boldsymbol{y} - \boldsymbol{x}\|_p}.$$

Intuitively, $L_p(\boldsymbol{x})$ characterizes the tightest Lipschitz bound in an arbitrarily small neighborhood of $\boldsymbol{x}$. If $f$ is differentiable at $\boldsymbol{x}$, then

$$L_p(\boldsymbol{x}) = \|\boldsymbol{J}_f(\boldsymbol{x})\|_p,$$

where $\|\boldsymbol{J}_f(\boldsymbol{x})\|_p$ denotes $\ell_p$-induced operator norm of the Jacobian matrix.

We now relate the global Lipschitz constant of a function to its local Lipschitz constant, following a result from Hytönen et al. (2016).

**Lemma 1** (Lipschitz constant via the Jacobian). *Let $f : \mathbb{R}^n \to \mathbb{R}^m$ be continuously differentiable. Then, for $1 \leq p \leq \infty$, the global Lipschitz constant of $f$ with respect to $\ell_p$ norm is given as*

$$L_p = \sup_{\boldsymbol{x} \in \mathbb{R}^n} L_p(\boldsymbol{x}) = \sup_{\boldsymbol{x} \in \mathbb{R}^n} \|\boldsymbol{J}_f(\boldsymbol{x})\|_p.$$

Thus, the global Lipschitz constant of a function can be expressed in a variational form involving the $\ell_p$-induced operator norm of its Jacobian matrix. To apply this principle to the softmax operator, we recall its Jacobian formulation from Gao & Pavel (2017), as stated below.

**Lemma 2** (Jacobian of the softmax). *Let $\sigma_\lambda : \mathbb{R}^n \to \Delta_n^\circ$ be the softmax function as in Definition 1 and let $s = \sigma_\lambda(x)$. Then, the Jacobian of $\sigma_\lambda$ at $x$ is*

$$J_{\sigma_\lambda}(x) = \lambda\big(\operatorname{Diag}(s) - ss^\top\big),$$

*so, in coordinates, if $s = \{s_1, s_2, \ldots s_n\}$,*

$$\big[J_{\sigma_\lambda}(x)\big]_{ij} = \begin{cases} \lambda\, s_i(1 - s_i), & i = j, \\ -\lambda\, s_i s_j, & i \neq j. \end{cases}$$

Refer to Proposition 2 in Gao & Pavel (2017) for the proof of Lemma 2. In Section 3, we will make use of Lemma 1 and the result in Lemma 2 for the softmax's Jacobian to establish our main results.

## 3 Main Results

We begin by deriving an inequality result for $\ell_p$-induced operator norms for matrices.

**Proposition 1** (Norm Interpolation). *Let $A \in \mathbb{R}^{n \times n}$. Then:*

*(a) $\|A\|_1 = \max_{1 \leq j \leq n} \sum_{i=1}^n |A_{ij}|$.*

*(b) $\|A\|_\infty = \max_{1 \leq i \leq n} \sum_{j=1}^n |A_{ij}|$.*

*(c) For $1 < p < \infty$,*

$$\|A\|_p \leq \|A\|_1^{1/p} \|A\|_\infty^{1-1/p}.$$

Proposition 1(a) and (b) state the well-known results that the $\ell_1$-induced operator norm of a matrix is equal to its maximum absolute column sum, and the $\ell_\infty$-induced operator norm is equal to the maximum absolute row sum (Golub & Van Loan, 2013). Proposition 1(c) establishes an interpolation inequality that upper bounds the $\ell_p$-induced operator norm of a matrix in terms of $\|A\|_1$ and $\|A\|_\infty$. While this inequality is a special case of the Riesz–Thorin interpolation theorem (Riesz, 1927; Stein, 1956; Thorin, 1939), we provide a self-contained proof in the Appendix A.1 for clarity. Importantly, Proposition 1(c) forms the key technical tool for this section, as it enables us to derive norm-uniform bounds on the Lipschitz constant of the softmax operator. In particular, we reformulate the optimization problem in Lemma 1 using the Jacobian form of the softmax provided in Lemma 2.

**Lemma 3** (Lipschitz constant of the softmax operator). *Let $\sigma_\lambda : \mathbb{R}^n \to \Delta_n^\circ$ be the softmax function as in Definition 1, and let $s = \sigma_\lambda(x)$. Then, for any $1 \leq p \leq \infty$, the Lipschitz constant of $\sigma_\lambda$ with respect to $\ell_p$ norm is given by*

$$L_p = \lambda \sup_{s \in \Delta_n^\circ} \big\| \operatorname{Diag}(s) - ss^\top \big\|_p. \tag{4}$$

The key implication of Lemma 3 is that the global Lipschitz constant reduces to the supremum of the induced matrix norm in Eq. 4 over the interior of the probability simplex. The boundary points of the simplex, $\partial\Delta_n$, do not influence the formulation of the Lipschitz constant, since all components of the softmax output are strictly positive.

Next, we combine the interpolation inequality from Proposition 1 with the Jacobian formulation of the softmax operator from Lemma 3 to establish its Lipschitz constant across all $\ell_p$ norms. This result is summarized as the main contribution of our work in Theorem 1.

**Theorem 1** (Lipschitz constant of softmax function). *Irrespective of the $\ell_p$ norms $(p \geq 1)$,*

*(a) $\|J_{\sigma_1}(x)\|_p \leq \frac{1}{2}$*

*(b) $\|\sigma_\lambda(x) - \sigma_\lambda(y)\|_p \leq \frac{\lambda}{2} \|x - y\|_p$*

The upper bound on the Jacobian of the softmax operator across $\ell_p$ norms, established in Theorem 1, directly provides the global Lipschitz constant of the softmax function. Refer to Appendix A.2 for a detailed proof. A natural question arises that while many works assume the constant to be 1, we show it is in fact $1/2$; but could it be even smaller, for instance $1/4$ as claimed by Xu et al. (2022). Proposition 2 resolves this question by proving that $\lambda/2$ is indeed the tight global Lipschitz constant of the softmax operator for any $\lambda > 0$.

**Proposition 2** (Tightness of the Lipschitz constant for softmax). *Consider the optimization problem,*

$$\sup_{\boldsymbol{x}\in\mathbb{R}^n} \|\boldsymbol{J}_{\sigma_1}(\boldsymbol{x})\|_p = \sup_{\boldsymbol{s}\in\Delta_n^{\circ}} \|\operatorname{Diag}(\boldsymbol{s}) - \boldsymbol{s}\boldsymbol{s}^{\top}\|_p,$$

*where $\Delta_n^{\circ}$ is the interior of the probability simplex.*

*(a) For $p = 1$ or $p = \infty$, the supremum is $1/2$ and is attained at an interior point in $\Delta_n^{\circ}$. In particular, for $\boldsymbol{x} = \big(\ln(n-1),\, 0,\, 0,\, \dots,\, 0\big) \in \mathbb{R}^n$, the softmax output $\boldsymbol{s} = \sigma_1(\boldsymbol{x})$ satisfies $\boldsymbol{s} \in \Delta_n^{\circ}$ and*

$$\|\operatorname{Diag}(\boldsymbol{s}) - \boldsymbol{s}\boldsymbol{s}^{\top}\|_p = \tfrac{1}{2}.$$

*(b) For $p \in (1, \infty)$, the supremum value is again $1/2$ but is not attained in $\Delta_n^{\circ}$ for $n > 2$. Instead, there exists a sequence $\{\boldsymbol{s}_k\}_{k\geq 1} \subset \Delta_n^{\circ}$ converging to the boundary of the probability simplex $\partial\Delta_n$ such that*

$$\lim_{k\to\infty} \|\operatorname{Diag}(\boldsymbol{s}_k) - \boldsymbol{s}_k\boldsymbol{s}_k^{\top}\|_p = \tfrac{1}{2},$$

*with $\boldsymbol{s}_k = \sigma_1(\boldsymbol{x}_k)$ for some $\boldsymbol{x}_k \in \mathbb{R}^n$. For $n = 2$, the supremum $\tfrac{1}{2}$ is attained at a point in $\Delta_2^{\circ}$.*

The proof of Proposition 2 is deferred to Appendix A.3. In particular, part (b) relies on the fact that for $1 < p < \infty$, the interpolation inequality in Proposition 1 is indeed strict for $\boldsymbol{J}_{\sigma_1}(\boldsymbol{x})$ for all $\boldsymbol{x} \in \mathbb{R}^n$. This is because the supremum in the optimization problem of Proposition 2 can be attained only on the boundary points of the probability simplex. Specifically, the extremal points are the permutations of $(1/2,\, 1/2,\, 0,\, \dots,\, 0)$, which lie in $\partial\Delta_n$.

*Remark.* There are multiple works showing that the Lipschitz constant of the softmax operator with respect to the $\ell_2$ norm is upper bounded by $1/2$. Alghamdi et al. (2022) upper bounds $\|\boldsymbol{J}_{\sigma_1}(\boldsymbol{x})\|_2$ by first controlling the Frobenius norm of the softmax Jacobian (using $\|\boldsymbol{A}\|_2 \leq \|\boldsymbol{A}\|_F$) and then maximising this bound over the simplex to obtain the same constant $1/2$. Yudin et al. (2025) analyse the Jacobian matrix $\boldsymbol{M}(\boldsymbol{s}) = \operatorname{diag}(\boldsymbol{s}) - \boldsymbol{s}\boldsymbol{s}^{\top}$ on the simplex and upper bound its spectral norm via an eigenvalue argument (reducing the worst case to a distribution supported on two classes), yielding $\|\boldsymbol{M}(\boldsymbol{s})\|_2 \leq 1/2$. Similarly, Newhouse (2025) gives an independent proof by applying Gershgorin's circle theorem to $\operatorname{diag}(\boldsymbol{s}) - \boldsymbol{s}\boldsymbol{s}^{\top}$, and also shows tightness in the $\ell_2$ setting. In contrast, we prove that the bound $1/2$ is tight uniformly across all $\ell_p$ norms for $p \geq 1$, and give a detailed analysis of attainability on the simplex for different $p$.

We next provide an example of a pair of points for which the empirically computed Lipschitz ratio approaches $1/2$. This gives empirical evidence for the tightness of our theoretical bound.

**Example 1** (Tightness of the Lipschitz bound). *Let $K = 20$ and $\varepsilon = 10^{-4}$. Consider the vectors*

$$\boldsymbol{x} = (0,\, 0,\, -K,\, -K,\, \dots,\, -K) \in \mathbb{R}^{10}, \qquad \boldsymbol{y} = \boldsymbol{x} + \varepsilon\,\boldsymbol{v},$$

*where $\boldsymbol{v}$ is eigenvector corresponding to the maximum eigenvalue of $J_{\sigma_1}(\boldsymbol{x})$. Then the ratio*

$$\frac{\|\sigma_1(\boldsymbol{y}) - \sigma_1(\boldsymbol{x})\|_p}{\|\boldsymbol{y} - \boldsymbol{x}\|_p}$$

*evaluates approximately to $0.49999999504472$ for all $p >= 1$. This demonstrates a concrete pair $(\boldsymbol{x}, \boldsymbol{y})$ where the Lipschitz constant of the softmax function is nearly attained. We can extend the example to any dimension by adding $-K$ as a value to the other added dimensions.*

# 4 Implications of the Improved Softmax Lipschitz Constant

Our result in Theorem 1 establishes that the softmax function is $\lambda/2$-Lipschitz with respect to all $\ell_p$ norms for $p \geq 1$. This tighter characterization enables us to revisit existing results where softmax Lipschitz continuity is relevant, leading to sharper constants or simplified identities. Next, we illustrate this in a few representative settings.

## 4.1 Refinement of (Gao & Pavel, 2017, Cor. 3).

Leveraging our sharper Lipschitz estimate, Corollary 3 in Gao & Pavel (2017) can be strengthened as follows.

**Corollary 1** (Softmax regularity with improved constants). *For any $\lambda > 0$, the softmax map $\sigma_\lambda : \mathbb{R}^n \to \Delta_n^\circ$ satisfies*

$$\text{(Lipschitz)} \quad \|\sigma_\lambda(\boldsymbol{x}) - \sigma_\lambda(\boldsymbol{y})\|_p \ \leq \ \tfrac{\lambda}{2} \|\boldsymbol{x} - \boldsymbol{y}\|_p, \qquad \forall \, p \geq 1,$$

$$\text{(co-coercive)} \quad \langle \sigma_\lambda(\boldsymbol{x}) - \sigma_\lambda(\boldsymbol{y}), \, \boldsymbol{x} - \boldsymbol{y} \rangle \ \geq \ \tfrac{2}{\lambda} \|\sigma_\lambda(\boldsymbol{x}) - \sigma_\lambda(\boldsymbol{y})\|_2^2.$$

*In particular,*

- $\sigma_\lambda$ *is nonexpansive and firmly nonexpansive for $\lambda \in (0, 2]$;*

- $\sigma_\lambda$ *is contractive for $\lambda \in (0, 2)$.*

The proof of this corollary follows directly from the analysis in Gao & Pavel (2017), with the Lipschitz constant of the softmax operator replaced by the sharper value $\lambda/2$.

## 4.2 Lipschitz Analysis of Attention variant

The self-attention module (Vaswani et al., 2017) refines an input feature matrix $\boldsymbol{X} \in \mathbb{R}^{n \times d}$ by projecting it into queries, keys, and values via learnable matrices $\boldsymbol{W}^Q \in \mathbb{R}^{d \times d_k}$, $\boldsymbol{W}^K \in \mathbb{R}^{d \times d_k}$, and $\boldsymbol{W}^V \in \mathbb{R}^{d \times d_v}$. The attention operation is then given by

$$\text{Att}(\boldsymbol{X}; \boldsymbol{W}^Q, \boldsymbol{W}^K, \boldsymbol{W}^V) = \text{softmax}\Big( \frac{\boldsymbol{X}\boldsymbol{W}^Q (\boldsymbol{X}\boldsymbol{W}^K)^\top}{\sqrt{d_k}} \Big) \boldsymbol{X}\boldsymbol{W}^V, \tag{5}$$

where softmax$(\cdot)$ denotes the row-wise application of the function $\sigma_1$. Although we obtain a tight global Lipschitz constant for the softmax operator $\sigma_1$ acting on the rows of the attention score matrix, the self-attention map $\boldsymbol{X} \mapsto \text{Att}(\boldsymbol{X}; \boldsymbol{W}^Q, \boldsymbol{W}^K, \boldsymbol{W}^V)$ defined on an unbounded input domain is not globally Lipschitz. Kim et al. (Kim et al., 2021) show that, for this map, the Lipschitz constant with respect to any $\ell_p$ norm is infinite. Motivated by this negative result, Qi et al. (2023) propose *scaled cosine similarity attention* (SCSA) as a Lipschitz-controlled variant of self-attention, which normalizes the projections $\boldsymbol{X}\boldsymbol{W}^Q$ and $\boldsymbol{X}\boldsymbol{W}^K$ to unit norm and applies an inverse temperature coefficient $\tau > 0$:

$$\text{SCSA}(\boldsymbol{X}; \boldsymbol{W}^Q, \boldsymbol{W}^K, \boldsymbol{W}^V, \nu, \tau) = \nu \, \text{softmax}\Big( \tau \frac{\boldsymbol{X}\boldsymbol{W}^Q (\boldsymbol{X}\boldsymbol{W}^K)^\top}{\|\boldsymbol{X}\boldsymbol{W}^Q\| \|\boldsymbol{X}\boldsymbol{W}^K\|} \Big) \boldsymbol{V}. \tag{6}$$

SCSA can be viewed as a Lipschitz variant of the standard attention mechanism. Theorem 1 in Qi et al. (2023) establishes an Lipschitz bound for scaled cosine similarity attention (SCSA), but their bound depends on the choice $(n-1)/n$ as a Lipschitz constant for the softmax operator. Using our sharper analysis of the softmax Lipschitz constant, we can replace $(n-1)/n$ by $1/2$ in their argument. Thus Lipschitz bound for SCSA can be tightened as shown below.

**Theorem 2** (Refined $\ell_2$-Lipschitz bound for SCSA). *Let $L_2(\text{SCSA})$ denote the global Lipschitz constant of $\boldsymbol{X} \mapsto \text{SCSA}(\boldsymbol{X}, \boldsymbol{W}^Q, \boldsymbol{W}^K, \boldsymbol{W}^V, \nu, \tau)$ with respect to the $\ell_2$ norm. Then, for fixed $(\boldsymbol{W}^Q, \boldsymbol{W}^K, \boldsymbol{W}^V, \nu, \tau)$,*

$$L_2(\text{SCSA}) \ \leq \ n^2 \, \nu \, \tau \, \varepsilon^{-1/2} \|\boldsymbol{W}^K\|_2 + n \, \nu \, \tau \, \varepsilon^{-1/2} \|\boldsymbol{W}^Q\|_2 + 2n \, \nu \, \varepsilon^{-1/2} \|\boldsymbol{W}^{V^\top}\|_2.$$

In particular, compared with the Lipschitz bound stated in Theorem 1 of Qi et al. (2023), Theorem 2 improves the estimate by a factor of 2, removing the extra factor 2 that appears in their original bound. Such refinements can directly impact theoretical robustness guarantees for attention-based architectures.

### 4.3 Entropy-regularized zero-sum games via double-softmax fixed point

Consider a two-player zero-sum matrix game with a payoff matrix $\boldsymbol{A} \in \mathbb{R}^{n \times m}$. The standard minimax formulation is

$$\min_{\boldsymbol{x} \in \Delta_n} \max_{\boldsymbol{y} \in \Delta_m} \boldsymbol{x}^\top \boldsymbol{A} \boldsymbol{y},$$

where $\Delta_n, \Delta_m$ denote the probability simplices. This formulation characterizes the Nash equilibrium of the game, capturing the row player's strategy that minimizes the worst-case loss against an adversarial opponent, and thus serves as a foundation for robust decision making, reinforcement learning, and adversarial training. To ensure unique mixed strategies and to improve algorithmic stability, it is common to add entropic regularization terms (Cen et al., 2021). The entropy-regularized problem becomes

$$\min_{\boldsymbol{x} \in \Delta_n} \max_{\boldsymbol{y} \in \Delta_m} \boldsymbol{x}^\top \boldsymbol{A} \boldsymbol{y} + \tau\big(H(\boldsymbol{x}) - H(\boldsymbol{y})\big),$$

where, $\tau > 0$ is the regularization parameter and $H(\cdot)$ is the Shannon entropy, defined for any probability vector $\boldsymbol{x} \in \Delta_n$ as $H(\boldsymbol{x}) = -\sum_{i=1}^{n} x_i \log x_i$.

A well-known algorithm to solve the above entropy regularization problem is the double-softmax fixed-point (DSFP) iteration (McKelvey & Palfrey, 1995), which updates

$$\boldsymbol{y}_{k+1} \leftarrow (1-\alpha)\,\boldsymbol{y}_k + \alpha\,\sigma_{1/\tau}\Big(\boldsymbol{A}^\top \sigma_{1/\tau}(-\boldsymbol{A}\boldsymbol{y}_k)\Big).$$

Intuitively, each player responds via a softmax update, yielding an equilibrium. Our improved Lipschitz constant for softmax in Theorem 1 provides a tighter condition to choose the regularization parameter $\tau$ depending on the payoff matrix $\boldsymbol{A}$, such that the DSFP iteration is contractive and convergent.

**Theorem 3** (Convergence of DSFP under entropy regularization). *Let $\boldsymbol{A} \in \mathbb{R}^{n \times m}$ be a payoff matrix and let $\tau > 0$ denote the entropy regularization parameter. For all $1 \le p \le \infty$, if $\tau > \|\boldsymbol{A}\|_p/2$ and $\alpha \in (0,1]$, then the DSFP iteration is a Banach contraction on $(\Delta_m, \|\cdot\|_p)$ with contraction factor $\|\boldsymbol{A}\|_p^2/(4\tau^2)$. Hence, the iterates $\boldsymbol{y}_k$ converge linearly in the $\ell_p$ norm to the unique fixed point $\boldsymbol{y}^\star \in \Delta_m$. The corresponding strategy of the row player is recovered as*

$$\boldsymbol{x}^\star = \sigma_{1/\tau}(-\boldsymbol{A}\boldsymbol{y}^\star),$$

*which is also convergent since $\boldsymbol{x}_k = \sigma_{1/\tau}(-\boldsymbol{A}\boldsymbol{y}_k)$ at each step.*

In summary, for DSFP iterations to converge linearly under the $\ell_p$ norm, we can derive a condition for the regularization parameter $\tau$ to satisfy and thereby guarantee well-defined equilibrium strategies.

## 5 Empirical Validation of Lipschitz Constant

In this section, we empirically validate our theoretical result that the softmax operator has a global Lipschitz constant of $1/2$, which is norm-uniform and tight. Our experiments involve a broad range of architectures across vision, language, and reinforcement learning, under varying datasets, prompts, and inverse temperature coefficient $\lambda$. All experiments were conducted on NVIDIA A40 GPUs. We compute the empirical Lipschitz constant using the definition in equation 3, using pairs of input obtained from each experimental setting. In particular, we compute,

$$\text{Empirical } L_p = \max_{1 \le i \le M} \frac{\|\sigma_\lambda(\boldsymbol{x}_i) - \sigma_\lambda(\boldsymbol{x}_i + \delta\boldsymbol{x}_i)\|_p}{\|\delta\boldsymbol{x}_i\|_p}, \tag{7}$$

where $M$ is the number of inputs. For each input sample $\boldsymbol{x}_i$, we apply small random perturbations $\delta\boldsymbol{x}_i$ normalized to have $\|\delta\boldsymbol{x}_i\|_p = \epsilon$, where $\epsilon$ denotes the perturbation magnitude. We report the computed Empirical $L_p$ constant across multiple values of $\epsilon$ and $p$.

## 5.1 Vision Models

We empirically evaluate the Lipschitz constant of the softmax operator, which appears in the attention mechanism of vision models. In particular, we consider Vision Transformer (ViT) models (Dosovitskiy et al., 2021) in three variants, Base (86M parameters), Large (307M parameters), and Huge (632M parameters). For each model, we select images from both the CIFAR-100 and ImageNet datasets, pass them through the transformer, and extract the pre-softmax attention scores $\boldsymbol{QK}^\top/\sqrt{d_k}$, which is the input to the softmax in Eq. 5. We then apply perturbations to these scores and compute the empirical Lipschitz constant row-wise according to Definition 7. $M$ is set to be 100 images.

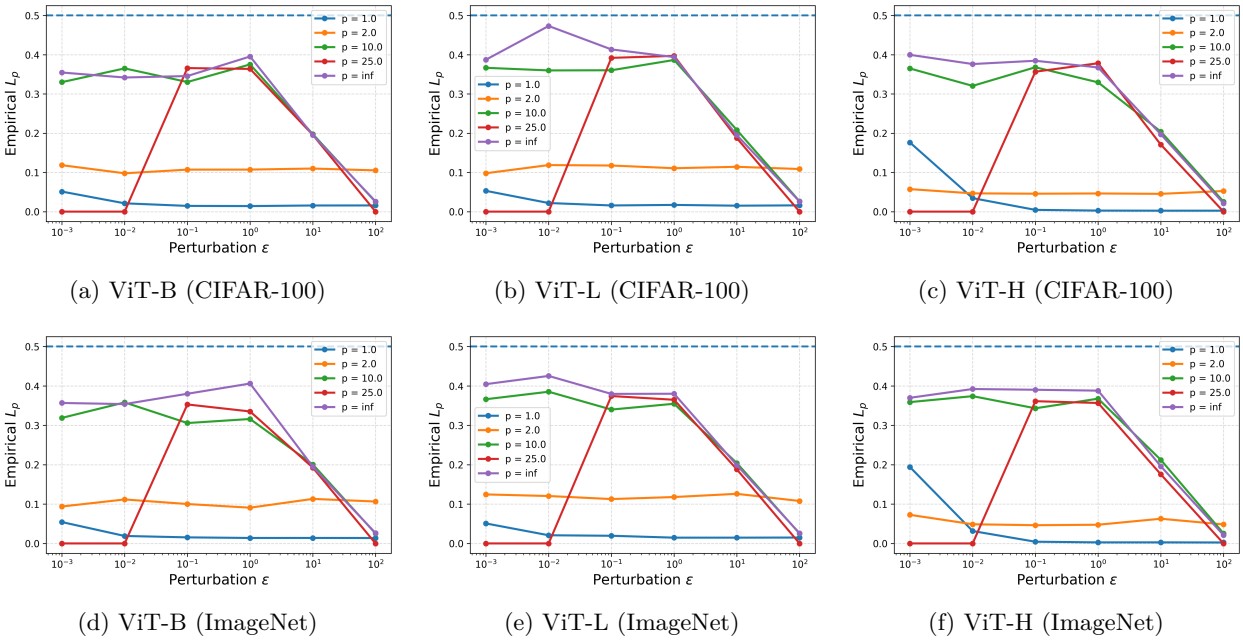

Figure 1: Empirical $L_p$ of the softmax operator over attention scores from three Vision Transformer (ViT) variants on CIFAR-100 (top row) and ImageNet (bottom row), across varying perturbation magnitudes $\epsilon$ for multiple $\ell_p$ norms. In all cases, the empirical values remain below the derived bound of $1/2$.

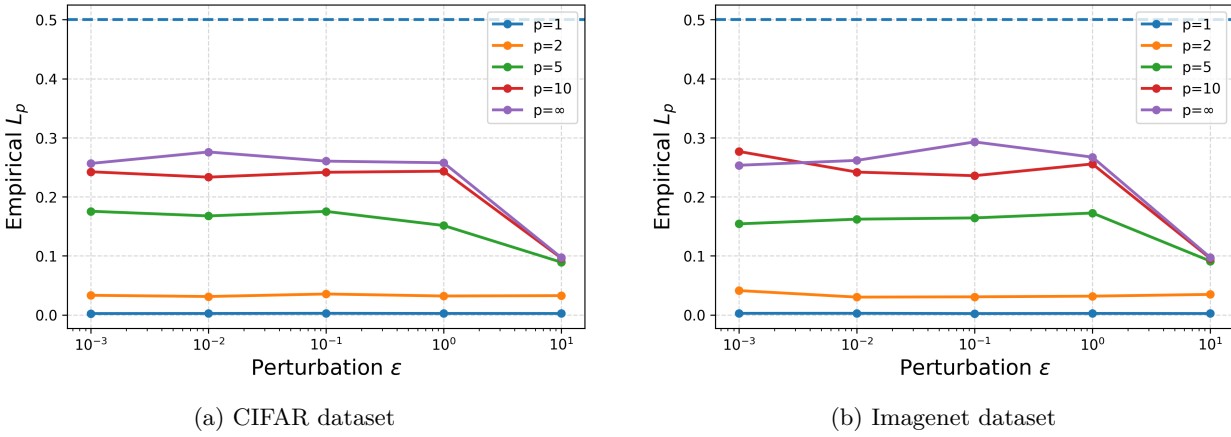

Figure 2: Empirical $L_p$ of the softmax operator for classification logits of ResNET50 model on (a) CIFAR-100 and (b) ImageNet dataset, across varying perturbation magnitudes $\epsilon$ for multiple $\ell_p$ norms. In all cases, the empirical values remain well below the derived bound of $1/2$.

The results are presented as plots of empirical Lipschitz values versus the perturbation norm in Fig. 1 for CIFAR-100 (top row) and ImageNet (bottom row) datasets, with the theoretical baseline of 0.5 indicated. We report results across multiple $\ell_p$ norms. In each case, the plotted value corresponds to the maximum empirical Lipschitz constant over all attention heads, layers, tokens, and images. In particular, the total number of tested vectors is significant, which is equal to $N \times H \times M \times 12$, where $N$ is the number of tokens and $H$ the number of heads. The plots consistently show that the empirical Lipschitz constant remains below our theoretical value of $1/2$, regardless of the choice of norm or ViT variant.

We further evaluate the Lipschitz constant of the softmax operator on the classification logits of ResNET50 model (He et al., 2016), using images from the CIFAR-100 dataset in Fig. 2a and ImageNet dataset in Fig. 2b. In this experiment, we use $M = 25,000$ images and introduce input perturbations to the logits and compute the empirical $L_p$ constant, for different $p$ and perturbation magnitude $\epsilon$. Consistent with our findings for attention scores, the empirical $L_p$ remains strictly below the theoretical bound of $1/2$.

## 5.2 Language Models

We next evaluate the Lipschitz constant of the softmax operator on attention scores of large language models. Specifically, we sample 100 prompts each from the HellaSwag (Zellers et al., 2019) and the PIQA (Bisk et al., 2020) datasets, process them through GPT-2 (Radford et al., 2019) (518M parameters) and Qwen3 (Bai et al., 2023) (8B parameters), extract the pre-softmax attention scores, apply perturbations, and compute the empirical Lipschitz constant as defined in Definition 7.

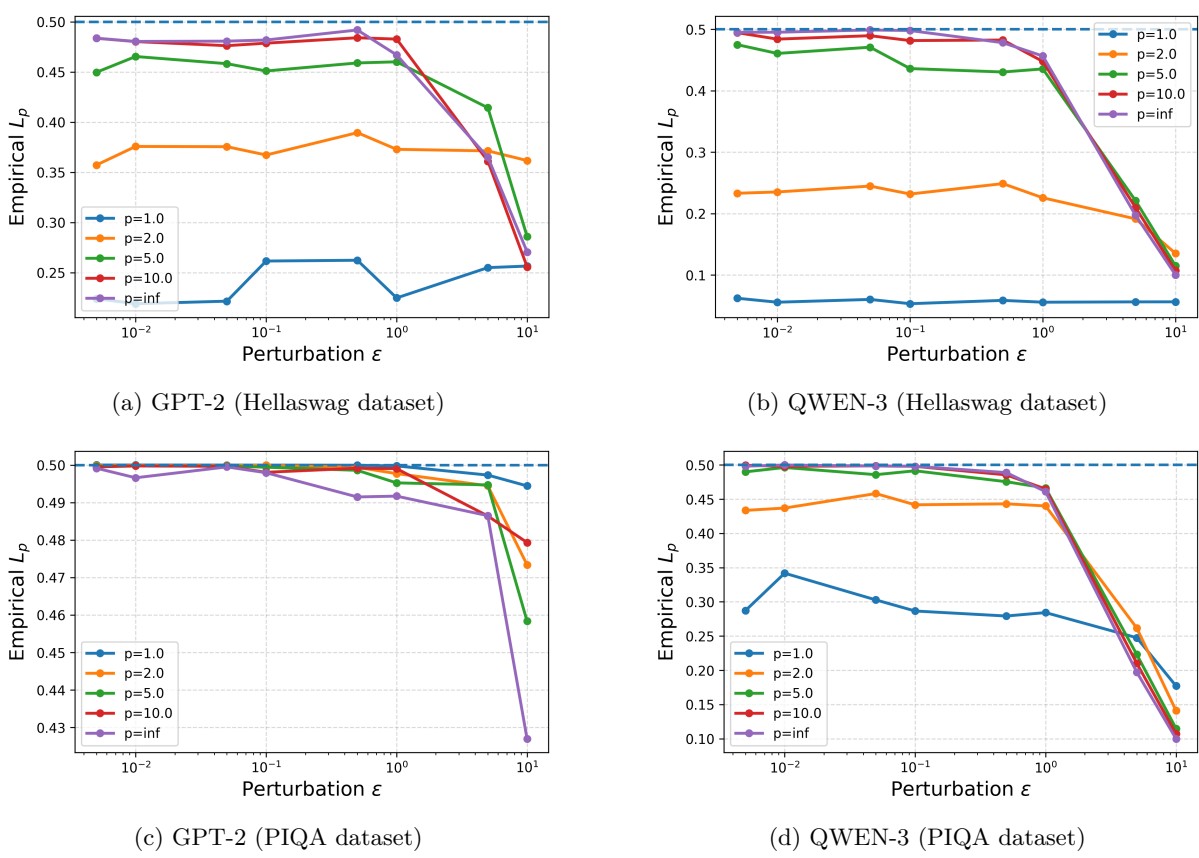

(a) GPT-2 (Hellaswag dataset)  (b) QWEN-3 (Hellaswag dataset)

(c) GPT-2 (PIQA dataset)  (d) QWEN-3 (PIQA dataset)

Figure 3: Empirical $L_p$ of the softmax operator over attention scores from two Large Language models, GPT-2 and QWEN-3, on Hellswag dataset (top row) and PIQA dataset (bottom row), across varying perturbation magnitudes $\epsilon$ for multiple $\ell_p$ norms. Across all configurations, the empirical values remain below the theoretical bound of $1/2$, with several instances approaching this limit, thereby confirming the tightness of the derived bound.

The scale of this evaluation is substantial here as well. The total number of tested vectors equals $H \times 100 \times 12$ per token per prompt, where the number of tokens varies with prompt length. Consistent with our experiments on vision models, the results in Fig. 3 for the Hellaswag dataset (top row) and the PIQA dataset (bottom row) demonstrate that the empirical Lipschitz constant remains strictly below the theoretical value of $1/2$ across all norms. Notably, for the PIQA dataset, we observe empirical constants of 0.4995 for the Qwen3 model and 0.4999 for GPT-2, providing empirical evidence for the tightness of our theoretically derived Lipschitz bound.

### 5.3 Reinforcement Learning Policies

We next investigate the Lipschitz behavior of softmax within stochastic policies in reinforcement learning. In policy-gradient methods (Sutton & Barto, 2018), a stochastic policy is typically parameterized as

$$\pi_\lambda(a \mid s) = \frac{\exp\big(\lambda Q(s, a)\big)}{\sum_{a' \in \mathcal{A}} \exp\big(\lambda Q(s, a')\big)},$$

where $Q(s, a)$ are the action logits for state $s \in \mathcal{S}$, $\mathcal{A}$ is the action space, and $\lambda > 0$ is the inverse temperature coefficient. We evaluate this mapping $Q(s, \cdot) \mapsto \pi_\lambda(\cdot \mid s)$ in the `LunarLander` environment (discrete action space of size $|\mathcal{A}| = 4$) and the `CartPole` environment ($|\mathcal{A}| = 2$) from the OpenAI Gym benchmark (Brockman et al., 2016). We implement a PPO (Proximal Policy Optimization) agent (Schulman et al., 2017) using the `stable_baselines3` library, initialized randomly, and executed to collect a large set of visited states. For each state, we perturb the corresponding action logits $Q(s, \cdot)$ with random perturbations, and compute the empirical Lipschitz constant as in Definition 7. The reported value is obtained by averaging over multiple states ($M = 25000$) and perturbation trials for different coefficients $\lambda$. The results in Fig. 4 for Cartpole and Lunarlander environments confirm that the empirical constants scale proportionally to $\lambda/2$, in agreement with our theoretical prediction. Note that for the `CartPole` environment, where the softmax input dimension is $n = 2$, we obtain an empirical constant of 1.9996 for $p = 8$ and $\lambda = 4.0$, which is very close to the derived bound.

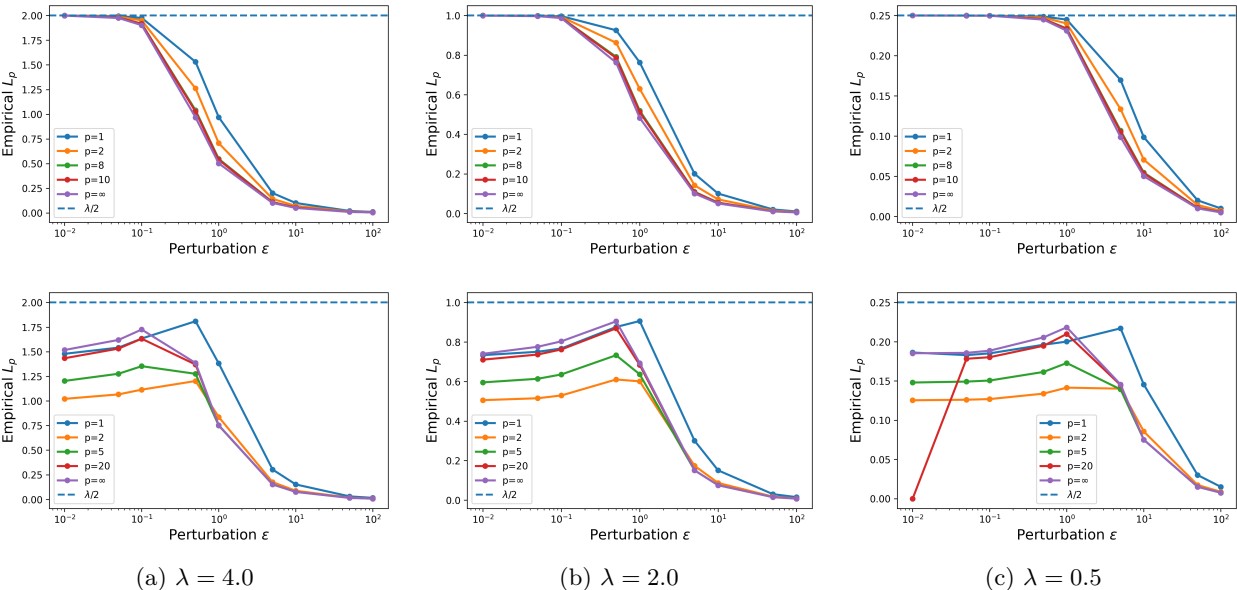

(a) $\lambda = 4.0$      (b) $\lambda = 2.0$      (c) $\lambda = 0.5$

Figure 4: Empirical Lipschitz sensitivity of the softmax policy in RL environments, Cartpole (top row) and Lunarlander (bottom row), across varying perturbation magnitudes $\epsilon$, under varying coefficients $\lambda$ for different $p$-norms. The empirical values scale with $\lambda$ and remain below the theoretical bound of $\lambda/2$, thereby empirically confirming the derived bound.

In summary, across all models, datasets, and settings considered, the empirical Lipschitz constant of softmax never exceeded $1/2$, thereby providing strong empirical validation of our theoretical bound. Furthermore, in

several cases, the empirical estimates approached 1/2, illustrating that the derived bound is indeed tight in practice.

## Conclusion

In this work, we derived that the softmax function admits a Lipschitz constant of 1/2 uniformly across all $\ell_p$ norms. This result clarifies a commonly adopted assumption in the machine learning literature, where the Lipschitz constant of the softmax operator has often been considered as 1 with respect to the $\ell_2$ norm. We further prove the tightness of our derived bound and demonstrate its utility in strengthening existing theoretical analyses. We also estimate the empirical Lipschitz constants for softmax transformations applied to attention score matrices from transformer-based architectures such as ViT, GPT-2, and Qwen3, classification logits from ResNet-50, and stochastic policies in reinforcement learning agents. In all cases, the empirical estimates remain strictly below the theoretical bound of 1/2, thereby guaranteeing the tightness and generality of our results.

### Broader Impact Statement

This work is primarily theoretical. By itself, this result does not introduce new datasets, architectures, or applications, but it provides sharper analytical tools for reasoning about the stability and robustness of existing models. On the positive side, such tools can help practitioners design systems with better controlled sensitivity to perturbations, complementing existing Lipschitz-based approaches used to improve robustness and training stability in deep networks (Cisse et al., 2017)(Miyato et al., 2018). At the same time, any advance that makes robust and stable deployment easier can be dual-use - it may also strengthen models used in societally sensitive settings, including surveillance and biometric monitoring (Almeida et al., 2022). A further risk is overstatement or misinterpretation - guarantees for a single component can be misconstrued as end-to-end assurances of safety, fairness, or reliability, which could encourage premature use in high-stakes contexts (AI, 2023).

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

# A  Appendix

## A.1  Proof of Proposition 1

(a) and (b) follow from standard results in matrix analysis (Golub & Van Loan, 2013).

***Proof of Proposition 1(c).*** Let $\boldsymbol{A} \in \mathbb{R}^{n \times n}$. For $p = 1$ and $p = \infty$, the identity trivially satisfies, and hence fix $1 < p < \infty$. For any $\boldsymbol{x} \in \mathbb{R}^n$,

$$\|\boldsymbol{A}\boldsymbol{x}\|_p^p = \sum_{i=1}^n \big|(\boldsymbol{A}\boldsymbol{x})_i\big|^p,$$

where $(\boldsymbol{A}\boldsymbol{x})_i$ denotes $i^{th}$ element in $\boldsymbol{A}\boldsymbol{x}$. Then, for each $i$, the triangle inequality gives

$$\big|(\boldsymbol{A}\boldsymbol{x})_i\big| = \Big|\sum_{j=1}^n A_{ij}x_j\Big| \leq \sum_{j=1}^n |A_{ij}|\,|x_j|$$

By Holder's inequality, we obtain for each $i$,

$$\sum_{j=1}^n |A_{ij}|\,|x_j| \leq \Big(\sum_{j=1}^n |A_{ij}|\Big)^{1-1/p}\Big(\sum_{j=1}^n |A_{ij}|\,|x_j|^p\Big)^{1/p}.$$

Thus, we can bound $\|\boldsymbol{A}\boldsymbol{x}\|_p^p$ as follows,

$$
\begin{aligned}
\|\boldsymbol{A}\boldsymbol{x}\|_p^p &\leq \sum_{i=1}^n \Big(\sum_{j=1}^n |A_{ij}|\Big)^{p-1}\Big(\sum_{j=1}^n |A_{ij}|\,|x_j|^p\Big) \\
&\leq \Big(\max_{1\leq i\leq n}\sum_{j=1}^n |A_{ij}|\Big)^{p-1}\sum_{i=1}^n\sum_{j=1}^n |A_{ij}|\,|x_j|^p \\
&= \|\boldsymbol{A}\|_\infty^{p-1}\sum_{j=1}^n |x_j|^p\Big(\sum_{i=1}^n |A_{ij}|\Big) \\
&\leq \|\boldsymbol{A}\|_\infty^{p-1}\Big(\max_{1\leq j\leq n}\sum_{i=1}^n |A_{ij}|\Big)\sum_{j=1}^n |x_j|^p \\
&= \|\boldsymbol{A}\|_\infty^{p-1}\,\|\boldsymbol{A}\|_1\,\|x\|_p^p.
\end{aligned}
$$

The second and fourth inequalities are obtained by taking the max term out of the summation[1], i.e,

$$\sum_{j=1}^n |A_{ij}| \leq \max_{1\leq i\leq n}\sum_{j=1}^n |A_{ij}| \text{ for all } i, \qquad \sum_{i=1}^n |A_{ij}| \leq \max_{1\leq j\leq n}\sum_{i=1}^n |A_{ij}| \text{ for all } j \tag{8}$$

The third and fifth equalities follow from $(a)$ and $(b)$. Taking the $p$-th root and maximizing over $\|\boldsymbol{x}\|_p = 1$, we get the inequality

$$\|\boldsymbol{A}\|_p \;\leq\; \|\boldsymbol{A}\|_\infty^{1-1/p}\,\|\boldsymbol{A}\|_1^{1/p}.$$

$\square$

---

[1]This inequality will be used in the proof of Proposition 2 as well.

### A.2 Proof of Theorem 1

***Proof of Theorem 1(a).*** Note that $\boldsymbol{J}_{\sigma_1}(x)$ is symmetric for all $\boldsymbol{x} \in \mathbb{R}^n$. Hence, from Proposition 1(a) and (b), we get that for all $\boldsymbol{x} \in \mathbb{R}^n$,

$$\|\boldsymbol{J}_{\sigma_1}(\boldsymbol{x})\|_1 = \|\boldsymbol{J}_{\sigma_1}(\boldsymbol{x})\|_\infty = \max_{1 \leq i \leq n} \sum_{j=1}^n |[\boldsymbol{J}_{\sigma_1}(x)]_{ij}|$$

From Lemma 2, the above optimization problem becomes,

$$\max_{1 \leq i \leq n} \; s_i(1 - s_i) + \sum_{j \neq i} s_i s_j$$

$$= \max_{1 \leq i \leq n} \; s_i(1 - s_i) - s_i^2 + \sum_{j=1}^n s_i s_j$$

$$= \max_{1 \leq i \leq n} \; 2s_i(1 - s_i),$$

where $\boldsymbol{s} = \sigma_1(\boldsymbol{x}) = (s_1, s_2, \ldots, s_n)$. The optimal value for the above problem is upper bounded by $1/2$, and $1/2$ is attained for all $\boldsymbol{x} \in \mathbb{R}^n$ such that the corresponding output $\boldsymbol{s} = \sigma_1(\boldsymbol{x})$ belongs to $\Delta_n^0$ and $s_i = 1/2$ for any $i \in \{1, 2, \ldots, n\}$. Thus, we get the following bounds,

$$\|\boldsymbol{J}_{\sigma_1}(\boldsymbol{x})\|_1 \leq \tfrac{1}{2} \quad \text{and} \quad \|\boldsymbol{J}_{\sigma_1}(\boldsymbol{x})\|_\infty \leq \tfrac{1}{2}$$

By Proposition 1(c), we get

$$\|\boldsymbol{J}_{\sigma_1}(\boldsymbol{x})\|_p \; \leq \; \left(\tfrac{1}{2}\right)^{\frac{1}{p}} \left(\tfrac{1}{2}\right)^{1 - \frac{1}{p}} = \tfrac{1}{2}, \quad \text{for all } 1 \leq p \leq \infty.$$

$\square$

***Proof of Theorem 1(b).*** From Theorem 1(a), we get,

$$\sup_{x \in \mathbb{R}^n} \|\boldsymbol{J}_{\sigma_1}(\boldsymbol{x})\|_p \; \leq \; \tfrac{1}{2}, \quad \text{for all } 1 \leq p \leq \infty.$$

From Lemma 3(b), for any $\lambda > 0$, we have,

$$\sup_{x \in \mathbb{R}^n} \|\boldsymbol{J}_{\sigma_\lambda}(\boldsymbol{x})\|_p \; \leq \; \frac{\lambda}{2} \quad \text{for all } 1 < p < \infty.$$

and hence using Lipschitz characterization in Lemma 1,

$$\|\sigma_\lambda(\boldsymbol{x}) - \sigma_\lambda(\boldsymbol{y})\|_p \; \leq \; \tfrac{\lambda}{2} \|\boldsymbol{x} - \boldsymbol{y}\|_p \quad \text{for all } \boldsymbol{x}, \boldsymbol{y} \in \mathbb{R}^n \text{ and for all } 1 \leq p \leq \infty.$$

$\square$

### A.3 Proof of Proposition 2

Define $\boldsymbol{M}(s) \in \mathbb{R}^{n \times n}$ as $\text{diag}(\boldsymbol{s}) - \boldsymbol{s}\boldsymbol{s}^T$. From Lemma 3, for all $1 \leq p \leq \infty$,

$$\sup_{\boldsymbol{x} \in \mathbb{R}^n} \|\boldsymbol{J}_{\sigma_1}(\boldsymbol{x})\|_p = \sup_{\boldsymbol{s} \in \Delta_n^\circ} \|\boldsymbol{M}(\boldsymbol{s})\|_p$$

***Proof of Proposition 2(a).*** As discussed in Theorem 1(a), for $p = 1$ and $p = \infty$, for all $\boldsymbol{s} \in \Delta_n^\circ$,

$$\|\boldsymbol{M}(\boldsymbol{s})\|_1 = \|\boldsymbol{M}(\boldsymbol{s})\|_\infty = \max_{1 \leq i \leq n} 2s_i(1 - s_i) \leq 1/2$$

and $1/2$ is attained if there exists $s \in \Delta_n^\circ$ such that $s_i = 1/2$ for some $i$ and $s_j > 0$ for $j \neq i$.

To prove that $1/2$ is indeed attained, we need to show an example where $1/2$ is attained at a point $\boldsymbol{s}$ in the interior of the probability simplex $\Delta_n^\circ$.

Let $\boldsymbol{x} \in \mathbb{R}^n$ such that $x_i = \ln(n-1)$ and $x_j = 0$ for all $j \neq i$, $\boldsymbol{s} = \sigma_1(\boldsymbol{x})$ is such that $s_i = 1/2$ and $s_j > 0$ for all $j \neq i$ and hence $\boldsymbol{s} \in \Delta_n^\circ$. Thus,

$$\max_{\boldsymbol{x} \in \mathbb{R}^n} \|\boldsymbol{J}_{\sigma_1}(\boldsymbol{x})\|_1 = \max_{\boldsymbol{x} \in \mathbb{R}^n} \|\boldsymbol{J}_{\sigma_1}(\boldsymbol{x})\|_\infty = 1/2$$

$\square$

***Proof of Proposition 2(b)***. We first prove for the case $n > 2$. It suffices to show that for arbitrary $1 < p < \infty$, $\|\boldsymbol{M}(\boldsymbol{s})\|_p \neq 1/2$ for any $\boldsymbol{s} \in \Delta_n^\circ$ and there exists $\boldsymbol{s} \in \partial\Delta_n$ such that $\|\boldsymbol{M}(\boldsymbol{s})\|_p = 1/2$ for $n > 2$ to prove that $1/2$ is indeed approached in limit. We need the following definition of the support of a vector and the corresponding lemma to prove this.

*Definition* A.1 (Support of a vector). For a vector $\boldsymbol{s} \in \mathbb{R}^n$, the support of $\boldsymbol{s}$ is defined as

$$\operatorname{supp}(\boldsymbol{s}) := \{ i \in \{1, \ldots, n\} : s_i \neq 0 \}.$$

**Lemma 4** (Support condition for attaining $1/2$). *Let $\boldsymbol{M}(\boldsymbol{s}) \in \mathbb{R}^{n \times n}$ be defined by*

$$\boldsymbol{M}(\boldsymbol{s}) := \operatorname{Diag}(\boldsymbol{s}) - \boldsymbol{s}\boldsymbol{s}^\top,$$

*for $\boldsymbol{s} \in \Delta_n$. Suppose $1 < p < \infty$ and $n > 2$. Then, $\|\boldsymbol{M}(\boldsymbol{s})\|_p = \frac{1}{2}$ if and only if $\boldsymbol{s}$ is a permutation of $(1/2, 1/2, 0, \ldots, 0)$*

The proof of the lemma is given towards the end of this section. From Lemma 4, there exists no $\boldsymbol{s} \in \Delta_n$ with $\operatorname{supp}(\boldsymbol{s}) > 2$ with $\|\boldsymbol{M}(\boldsymbol{s})\|_p = 1/2$. Hence, there exists no $\boldsymbol{s} \in \Delta_n^\circ$ with $\|\boldsymbol{M}(\boldsymbol{s})\|_p = 1/2$ since $\operatorname{supp}(\boldsymbol{s}) = n$ for all $\boldsymbol{s} \in \Delta_n^\circ$. Also, there exists $\boldsymbol{s}$ with $\operatorname{supp}(\boldsymbol{s}) = 2$ such that $\|\boldsymbol{M}(\boldsymbol{s})\|_p = 1/2$.

Let $\hat{\boldsymbol{s}} \in \Delta_n$ with $\operatorname{supp}(\hat{\boldsymbol{s}}) = 2$ be such that $\|\boldsymbol{M}(\hat{\boldsymbol{s}})\|_p = 1/2$. By definition, $\hat{\boldsymbol{s}} \in \partial\Delta_n$ (boundary of the simplex). Hence, we can obtain a sequence of $\boldsymbol{s}_k \in \Delta_n^\circ$ (interior of the simplex) such that $\boldsymbol{s}_k \to \hat{\boldsymbol{s}}$ as $k \to \infty$ (Rockafellar & Wets, 1998). Also since $\boldsymbol{M}(\boldsymbol{s})$ is a continuous function, $\boldsymbol{M}(\boldsymbol{s}_k) \to 1/2$ as $k \to \infty$. For completeness, we show convergence by constructing example sequence in Example 2.

Finally, when $n = 2$, the point $\boldsymbol{s} = (1/2, 1/2)$ lies in $\Delta_2^\circ$ and attains $\|\boldsymbol{M}(\boldsymbol{s})\|_p = 1/2$. $\square$

**Example 2** (Sequence $(\boldsymbol{s}_k)_{k \geq 1} \subset \Delta_n^\circ$ such that $\boldsymbol{M}(\boldsymbol{s}_k) \to 1/2$ as $k \to \infty$). *Consider $\hat{\boldsymbol{s}} \in \partial\Delta_n$ with $\hat{s}_1 = \hat{s}_2 = 1/2$ and all other entries zero. By Lemma 4, $\|\boldsymbol{M}(\hat{\boldsymbol{s}})\|_p = 1/2$ To approximate $\hat{\boldsymbol{s}}$ from the interior of the probability simplex, fix $\epsilon \in (0, 1/2)$ and choose $\delta \in (0, 1/2)$ satisfying $2\delta(1 - \delta) = \epsilon$. Define*

$$s_1 = s_2 = \tfrac{1}{2} - \delta, \qquad s_j = \tfrac{2\delta}{n-2}, \ \ j \geq 3,$$

*so that $\boldsymbol{s} \in \Delta_n^\circ$. Consider the vector*

$$\boldsymbol{v} = \left(2^{-1/p}, -2^{-1/p}, 0, \ldots, 0\right) \in \mathbb{R}^n \quad with \quad \|\boldsymbol{v}\|_p = 1,$$

*A direct computation shows,*

$$[\boldsymbol{M}(\boldsymbol{s})\boldsymbol{v}]_1 = -[\boldsymbol{M}(\boldsymbol{s})\boldsymbol{v}]_2 = \frac{2}{2^{1/p}}\left(\frac{1}{2} - \delta\right)^2 \ and \ [\boldsymbol{M}(\boldsymbol{s})\boldsymbol{v}]_j = 0 \ for \ all \ j \notin \{1, 2\}.$$

*Hence, we get,*

$$\|\boldsymbol{M}(\boldsymbol{s})\boldsymbol{v}\|_p = \tfrac{1}{2} - 2\delta(1 - \delta) = \tfrac{1}{2} - \epsilon.$$

*Thus for every $\epsilon \in (0, 1/2)$, there exists $\boldsymbol{s} \in \Delta_n^\circ$ with $1/2 - \epsilon \leq \|\boldsymbol{M}(\boldsymbol{s})\|_p < 1/2$, proving that $1/2$ is the tight upper bound.*

***Proof of Lemma 4.*** Let $s$ be a permutation of $(1/2, 1/2, 0, \ldots, 0)$ i.e $s_i = s_j = 1/2$ for some distinct $i, j \in \{1, 2, \ldots, n\}$. $\boldsymbol{M}(\boldsymbol{s})$ has values $\boldsymbol{M}(\boldsymbol{s})_{ii} = \boldsymbol{M}(\boldsymbol{s})_{jj} = 1/4$, $\boldsymbol{M}(\boldsymbol{s})_{ij} = \boldsymbol{M}(\boldsymbol{s})_{ji} = -1/4$, and zeros elsewhere. To show that $\|\boldsymbol{M}(\boldsymbol{s})\|_p = 1/2$, it suffices to construct $\boldsymbol{v} \in \mathbb{R}^n$ such that $\|\boldsymbol{v}\|_p = 1$ and $\|\boldsymbol{M}(\boldsymbol{s})\boldsymbol{v}\|_p = 1/2$. Consider $\boldsymbol{v}$ to be a similar permutation of

$$\left(2^{-1/p}, -2^{-1/p}, 0, \ldots, 0\right) \in \mathbb{R}^n.$$

i.e. $v_i = 2^{-1/p}$ and $v_j = -2^{-1/p}$. A direct calculation gives $\|\boldsymbol{M}(\boldsymbol{s})\boldsymbol{v}\|_p = 1/2$.

Now we need to prove that if $\|\boldsymbol{M}(\boldsymbol{s})\|_p = 1/2$, $\boldsymbol{s}$ is a permutation of $(1/2, 1/2, 0, \ldots, 0)$. By Proposition 1(c), the interpolation inequality gives

$$\|\boldsymbol{M}(\boldsymbol{s})\|_p \leq \|\boldsymbol{M}(\boldsymbol{s})\|_1^{1/p} \|\boldsymbol{M}(\boldsymbol{s})\|_\infty^{1-1/p}.$$

From Proposition 2(a), we know $\|\boldsymbol{M}(\boldsymbol{s})\|_1 = \|\boldsymbol{M}(\boldsymbol{s})\|_\infty \leq 1/2$. Thus if $\|\boldsymbol{M}(\boldsymbol{s})\|_p = 1/2$, equality must hold throughout. This means for all $\boldsymbol{s} \in \Delta_n$ such that $\|\boldsymbol{M}(\boldsymbol{s})\|_p = 1/2$, we have $\|\boldsymbol{M}(\boldsymbol{s})\|_1 = \|\boldsymbol{M}(\boldsymbol{s})\|_\infty = 1/2$.

We divide into cases according to the support size of $\boldsymbol{s}$.

**Case (i):** $\operatorname{supp}(\boldsymbol{s}) = 1$. Suppose $s_i = 1$ for some $i$, and $s_j = 0$ for all $j \neq i$. Then $\boldsymbol{M}(\boldsymbol{s}) = 0$. Hence $\|\boldsymbol{M}(\boldsymbol{s})\|_p = 0$ for all $p$. Thus, the value $1/2$ cannot be attained for any $\boldsymbol{s}$.

**Case (ii):** $\operatorname{supp}(\boldsymbol{s}) = 2$. As discussed in Theorem 1(a), for $p = 1$ and $p = \infty$, for all $\boldsymbol{s} \in \Delta_n$,

$$\|\boldsymbol{M}(\boldsymbol{s})\|_1 = \|\boldsymbol{M}(\boldsymbol{s})\|_\infty = \max_{1 \leq i \leq n} 2s_i(1 - s_i) \leq 1/2$$

and $1/2$ is attained only if there exists $s \in \Delta_n$ such that $s_i = 1/2$ for some $i$. Thus, if $\operatorname{supp}(\boldsymbol{s}) = 2$, then $\boldsymbol{s}$ needs to be permutations of $(1/2, 1/2, 0, \ldots, 0)$ for $\|\boldsymbol{M}(\boldsymbol{s})\|_1 = \|\boldsymbol{M}(\boldsymbol{s})\|_\infty = 1/2$ and we have already shown $\|\boldsymbol{M}(\boldsymbol{s})\|_p = 1/2$ for any permutation of $(1/2, 1/2, 0, \ldots 0)$.

Thus, if $\|\boldsymbol{M}(\boldsymbol{s})\|_p = 1/2$ with $\operatorname{supp}(\boldsymbol{s}) = 2$, then $\|\boldsymbol{M}(\boldsymbol{s})\|_1 = \|\boldsymbol{M}(\boldsymbol{s})\|_\infty = 1/2$ which needs $\boldsymbol{s}$ to be a permutation of $(1/2, 1/2, 0, \ldots 0)$.

**Case (iii):** $\operatorname{supp}(\boldsymbol{s}) > 2$. Suppose, $\hat{\boldsymbol{s}} \in \Delta_n$ with $\operatorname{supp}(\hat{\boldsymbol{s}}) > 2$ satisfies $\|\boldsymbol{M}(\hat{\boldsymbol{s}})\|_p = 1/2$. By Proposition 1(c),

$$\|\boldsymbol{M}(\hat{\boldsymbol{s}})\|_p \leq \|\boldsymbol{M}(\hat{\boldsymbol{s}})\|_1^{1/p} \|\boldsymbol{M}(\hat{\boldsymbol{s}})\|_\infty^{1-1/p}.$$

Since $\|\boldsymbol{M}(\hat{\boldsymbol{s}})\|_1 = \|\boldsymbol{M}(\hat{\boldsymbol{s}})\|_\infty = 1/2$ holds, equality must hold in the interpolation inequality in Proposition 1. Thus, it suffices to rule out equality.

We just need to show that one of the inequalities used in the proof of Proposition 1(c) is strict. We show that the inequality in Eq. 8 is strict i.e. there exists $1 \leq i \leq n$, such that,

$$\sum_{j=1}^n |\boldsymbol{M}(\boldsymbol{s})_{ij}| < \max_{1 \leq k \leq n} \sum_{j=1}^n |\boldsymbol{M}(\boldsymbol{s})_{kj}|$$

Since $\|\boldsymbol{M}(\hat{\boldsymbol{s}})\|_1 = \|\boldsymbol{M}(\hat{\boldsymbol{s}})\|_\infty = 1/2$, $\hat{s}_{i_1} = 1/2$ for some $1 \leq i_1 \leq n$ and $\hat{s}_j < 1/2$ for all $j \neq i_1$. Since $\operatorname{supp}(\hat{\boldsymbol{s}}) > 2$, there exists an index $i_2 \in \{1, 2, \ldots, n\}$ and $i_2 \neq i_1$, where $\hat{s}_{i_2} < 1/2$. As discussed in Theorem 1, for any row index $i$, $\sum_{j=1}^n |\boldsymbol{M}(\hat{\boldsymbol{s}})_{ij}| = 2\hat{s}_i(1 - \hat{s}_i)$. Hence,

$$\max_{1 \leq i \leq n} \sum_{j=1}^n |\boldsymbol{M}(\boldsymbol{s})_{ij}| = \sum_{j=1}^n |\boldsymbol{M}(\boldsymbol{s})_{i_1 j}| = 1/2,$$

and for row index $i_2$,

$$\sum_{j=1}^n |\boldsymbol{M}(\boldsymbol{s})_{i_2 j}| < 1/2 = \max_{1 \leq k \leq n} \sum_{j=1}^n |\boldsymbol{M}(\boldsymbol{s})_{kj}|.$$

This shows that the inequality in Eq. 8 is strict. Hence, the interpolation inequality in Proposition 1(c) is strict for the matrix $\boldsymbol{M}(\hat{\boldsymbol{s}})$. This contradicts our assumption that $\|\boldsymbol{M}(\hat{\boldsymbol{s}})\|_p = 1/2$.

Combining all cases, we conclude that $\|\boldsymbol{M}(\boldsymbol{s})\|_p = 1/2$ if and only if $\boldsymbol{s}$ is a permutation of $(1/2, 1/2, 0 \ldots 0)$.

$\square$

### A.4 Proof of Theorem 3

Let the DSFP map be defined as

$$T(\boldsymbol{y}) \;=\; \sigma_{1/\tau}\Big(\boldsymbol{A}^\top \sigma_{1/\tau}(-\boldsymbol{A}\boldsymbol{y})\Big), \qquad \boldsymbol{y} \in \Delta_m,$$

where $\sigma_{1/\tau}$ denotes the softmax operator with temperature $\tau$. By Theorem 1, the softmax operator is globally Lipschitz with constant $\frac{1}{2\tau}$, uniformly across all $\ell_p$ norms with $1 \le p \le \infty$. Applying the chain rule of Lipschitz constants, we obtain

$$\|T(\boldsymbol{y}_1) - T(\boldsymbol{y}_2)\|_p \;\le\; \frac{1}{2\tau}\,\|\boldsymbol{A}^\top\|_p\,\frac{1}{2\tau}\,\|\boldsymbol{A}\|_p\,\|\boldsymbol{y}_1 - \boldsymbol{y}_2\|_p.$$

Since $\|\boldsymbol{A}^\top\|_p = \|\boldsymbol{A}\|_p$, this yields

$$\|T(\boldsymbol{y}_1) - T(\boldsymbol{y}_2)\|_p \;\le\; \frac{\|\boldsymbol{A}\|_p^2}{4\tau^2}\,\|\boldsymbol{y}_1 - \boldsymbol{y}_2\|_p.$$

Hence, the DSFP map has the Lipschitz constant,

$$c \;=\; \frac{\|\boldsymbol{A}\|_p^2}{4\tau^2}.$$

If $\tau > \|\boldsymbol{A}\|_p/2$, then $c < 1$ and $T$ is a Banach contraction on the complete metric space $(\Delta_m, \|\cdot\|_p)$. By the Banach fixed-point theorem, $T$ then admits a unique fixed point $\boldsymbol{y}^\star \in \Delta_m$, and for any initialization $\boldsymbol{y}_0 \in \Delta_m$, the iteration

$$\boldsymbol{y}_{k+1} \;\leftarrow\; (1-\alpha)\,\boldsymbol{y}_k + \alpha\,T(\boldsymbol{y}_k), \qquad \alpha \in (0,1],$$

converges linearly in the $\ell_p$ norm to $\boldsymbol{y}^\star$. At each step, the row player's strategy is updated as

$$\boldsymbol{x}_k = \sigma_{1/\tau}(-\boldsymbol{A}\boldsymbol{y}_k).$$

Since $\boldsymbol{y}_k \to \boldsymbol{y}^\star$ and the softmax map is continuous, it follows that $\boldsymbol{x}_k \to \boldsymbol{x}^\star := \sigma_{1/\tau}(-\boldsymbol{A}\boldsymbol{y}^\star)$. Thus, both players' equilibrium strategies are recovered in the limit.

