# OpenReview forum: "Softmax is $1/2$-Lipschitz: A tight bound across all $\ell_p$ norms"
_TMLR — Accepted by TMLR_

### Review · Reviewer_1Ccc · 2025-11-12

**Summary Of Contributions:**

The paper derives a tight upper bound of the Lipschitz constant of the softmax operator. Various validations are conducted across model architectures (e.g., Qwen3, ViT) and tasks (e.g., Vision, RL, Text generation). However, I am not familiar with the theorem, and I do not verify the correctness of the theorem in the paper.

**Audience:**

Yes

**Audience Explanation:**

Lipschitz constant of the **complete neural network** is vital in the community, because it reflects the lower bound of the eigenvalues of the Hessian Matrix of the neural network and plays an important role in the analysis of the convergence in optimization. However, the Lipschitz constant of the softmax operator is less important than that of the complete neural network.

**Claims And Evidence:**

Yes

**Claims Explanation:**

The paper validates the $1/2$ Lipschitz constant of the softmax operator by the approximation of Lipschitz constant of the softmax operator across various model architectures and tasks.

**Requested Changes:**

My concerns focus on the potential application and inspiration of the proposed theorem.

1. About the application, as I pointed, the tight upper bound of the Lipschitz constant of the softmax operator cannot totally reflect the properties of the neural network in the optimization, because what we indeed need is the Lipschitz constant of the **complete neural network**, rather than the Lipschitz constant of a single operator. More discussions on the relation between the Lipschitz constant of the softmax and the Lipschitz constant of the neural network are expected.

2. Softmax operator usually attaches to the top/deep of the neural network, except for transformers. Can any normalization/regularization techniques be inspired by the theorem, e.g., Spectral Normalization in GANs? More discussions are expected. As for transformers, can the application of softmax operator in Multi Head Attention make the training of transformers faster and more stable?

3. The novelty and correctness of the theorem need to be verified by other reviewers and the editor.

---

> ### Author Response · Authors · 2025-11-25
> **Response to Reviewer 1Ccc**
>
> We thank the reviewer for the positive assessment of the paper's relevance and for the comment on applications and the relation of the Lipschitz constant of the softmax operator to the Lipschitz properties of complete neural networks.
>
> **1. About the application, as I pointed, the tight upper bound of the Lipschitz constant of the softmax operator cannot totally reflect the properties of the neural network in the optimization, because what we indeed need is the Lipschitz constant of the complete neural network, rather than the Lipschitz constant of a single operator. More discussions on the relation between the Lipschitz constant of the softmax and the Lipschitz constant of the neural network are expected.**
>
> We agree that the Lipschitz constant of the complete neural network is the quantity of primary interest in robustness and optimization analyses, and that the constant of a single operator, such as softmax, provides only one component of this picture. In many existing analyses, the overall network constant is indeed obtained by composing Lipschitz constants of individual layers. In those settings, replacing the commonly used bound $1$ for softmax by our tighter constant $1/2$ (uniform across all $\ell_{p}$ norms) directly tightens the resulting overall bound. This perspective is already reflected in our discussion in Section $4.2$, where we explain how our result yields a sharper approximation of the Lipschitz constant for a variant of self-attention.
>
> In the revised manuscript, we have strengthened the discussion of how our result interfaces with whole-network analyses. In particular, in the second paragraph of the Introduction (within the discussion on applications of an accurate softmax Lipschitz analysis), we have added a separate remark explaining that when a network is written as a composition of affine, activation, and softmax layers, standard arguments upper bound the global Lipschitz constant by an appropriate composition (often a product) of layer-wise Lipschitz constants. We also cite representative works to support the statement. Thus, replacing the commonly used constant with our tight value immediately yields sharper network-level or layer-level guarantees (refer to Theorem $2$ in the manuscript for an explicit example).
>
> **2. Softmax operator usually attaches to the top/deep of the neural network, except for transformers. Can any normalization/regularization techniques be inspired by the Theorem, e.g., Spectral Normalization in GANs? More discussions are expected. As for transformers, can the application of softmax operator in Multi Head Attention make the training of transformers faster and more stable?**
>
> We appreciate this suggestion.
>
> a) **Spectral Normalization in GANs**: When softmax is used only at the top of a network, our result implies that to obtain a $\beta$-Lipschitz model, one only needs the composition of remaining layers to be $2\beta$-Lipschitz; this could be exploited to increase the effective expressive capacity under Lipschitz constraints and, in turn, perform spectral normalization. We are working in this direction to obtain more empirical evidence.
>
> b) **Training more stable transformers**: We hypothesize that we can build a state-of-the-art Lipschitz-controlled and thereby robust and stable attention network by normalizing the attention layers with a better upper bound derived using the tighter Lipschitz constant. This is also a part of future work and requires empirical evidence.
>
> Since we do not yet have solid empirical evidence to substantiate these potential implications, we refrain from making claims beyond the theoretical refinements established in this work. Instead, in the second paragraph of the Introduction (immediately after defining softmax), we restrict our discussion to applications of softmax Lipschitz constants that are supported by existing published results.

---

> > ### Comment · Reviewer_1Ccc · 2025-11-26
> > **Replies to the Response of Authors**
> >
> > Thank the authors for their exquisite and thorough responses. My concerns are addressed.
> >
> > The topic of sharpest upper bound of Lipschitz Constant of the Neural Network or a single operator is vital to the machine learning community. The potential value of this work can be expected.
> >
> > A minor suggestion, for readers without too much theory backgrounds. Summaries like "TL; DR" are welcome for the further broadcast of such valuable works.

---

> > > ### Author Response · Authors · 2025-12-10
> > > **Reply to rebuttal response**
> > >
> > > Thank you for confirming that our responses addressed your concerns. As suggested, in the next revision, we will add a short TL;DR to the OpenReview page.

---

### Review · Reviewer_GXEw · 2025-11-15

**Summary Of Contributions:**

This paper demonstrates that the Lipschitz bound for the Softmax function is actually $\frac{\lambda}{2}$, where $\lambda$ is the temperature parameter, and that this bound is tight. The authors also run numerical experiments to support their derivations, and include examples to justify the usefulness of the Lipschitz bound.

**Strengths**
I believe the strengths of the paper are its clarity of presentation and the completeness of the authors’ analysis. Regarding clarity, the paper is well written with almost no grammar/spelling issues, it is nicely organized, and the math is logically and clearly presented. Regarding completeness, the authors provide nice proofs of their claims, a comprehensive numerical analysis, and some nice illustrative examples.

**Weaknesses**
The paper is well written and researched and I can only really find one weakness—and it should be easily resolvable. Specifically, the authors gloss over broader impacts. See broader impact concerns section for more details.

Beyond this, I have found a few small errors and have provided a few suggestions in the requested changes below.

**Audience:**

Yes

**Audience Explanation:**

Given the relevance of the softmax function to modern ML approaches the reviewer believes that this is relevant material for the TMLR community.

**Broader Impact Concerns:**

The paper argues for the impactfulness of the provided characterization of the softmax Lipschits constant—e.g. with respect to machine learning and game theory. Thus, the statement “There are no possible repercussions of this work. There is no potential negative impact that a user of this research should be aware of” seems somewhat contradictory. If this work is indeed relevant to machine learning and game theory communities then it would have both positive and negative societal impacts. It would be nice to mention the potential for at least some such impacts. For example, on the machine learning/AI side, the reviewer is aware of quite a bit of literature on the topic so this should not be too difficult to do. For easily digestible overviews of AI impacts one could look at the Atlas of AI by Kate Crawford or The AI Con by Hanna and Bender. For academic articles one could start with works by the same authors.

**Claims And Evidence:**

Yes

**Claims Explanation:**

As far as I can tell the mathematical statements are all either backed up with references or proofs, and the empirical evidence seems well presented. Note that I have verified the correctness of the appendices (except the construction of example convergent sequences on page 15, which was not required for proving the paper's results), but I have not reviewed any experimental code.

**Requested Changes:**

I list the requested changes below. I have added “**C**” at the end of changes that I view as critical, for my review, and “**S**” after ones that I view more as suggestions.

- [line 2 intro]: “logits is normalized” => “logits are normalized” **C**
- Proposition 1 (c): what is $t^0$? **C**
- [Example 2, page 5]: this is labelled “Example 2” but I’m pretty sure it’s the first example to appear? **C**
- [end of page 6]: “Such refinements are vital, as they directly impact theoretical robustness guarantees for attention-based architectures.” I wonder if this is a slight overstatement? Does removing a factor of $2$ here really provide any significant change in how people understand attention architectures? **S**
- [Fig 2]: minor issue, but it would be ideal to move figure 2 into the section for vision models, if it is possible to rearrange the paper layout in that way while not introducing wasted space. If not, then no problem **S**
- [A.2 4th line of text]: shouldn’t $x$ be in $\mathbb{R}^n$?
- [A.2 4th line of text]: should $\Delta_n^0$ be changed to $\Delta_n^{\circ}$ to keep notation consistent? **C**
- [appendix]: could be nice to add something at the end of proofs to better separate them. E.g., between proof of Thm 1 parts (a) and (b). **S**
- [proof of proposition 2(b) line 1]: perhaps change “It suffices to show that for any $1 < p < \infty, \dots$” to “It suffices to show that for arbitrary $1 < p < \infty, \dots$”. **S**
- [Lemma 4 Eq. 4]: should $j = \{1,2\}$ be instead $j \not\in \{1, 2\}$? **C**
- [end of page 15]: “proof of Lemma” => “proof of Lemma 4”.
- [second last text line of page 15]: “It” should be “it”
- [3rd last text line of page 15]: I think $M(s)_{ji}$ should be equal to $-\frac{1}{4}$.
- [Proof of Lemma, case (ii), text line 2, page 16]: suggest change “attained if” => “attained only if”. **S**
- [Last equation page 16]: redundancy: the exact same equation is repeated twice (I assume you meant to switch $i_2$ with $i_3$ in the second equation. Also, if I am not mistaken you don’t need to mention $i_3$ at all but can just say that $s_{i_2} < ½$, right? Might be useful to do this for conciseness **S**.
- Lastly, the reviewer is aware of a paper that shows that softmax self-attention is not Lipschitz (Hyunjik Kim, George Papamakarios, Andriy Mnih Proceedings of the 38th International Conference on Machine Learning, PMLR 139:5562-5571, 2021). Given how self-attention is one of the big modern applications of the softmax function for machine learning, it would be great to say a few words highlighting how the softmax can be Lipschitz while self attention is not.

---

> ### Author Response · Authors · 2025-11-25
> **Response to Reviewer GXEw**
>
> We thank the reviewer for the thorough, line-by-line feedback on both the technical content and the mathematical arguments; these comments have been extremely helpful in refining and improving the manuscript.
>
> **1. [line 2 intro]: logits is normalized $\to$ logits are normalized.**
>
>  Corrected
>
> **2. Proposition 1(c): what is the meaning of $t^0$?**
>
> Thank you for pointing out this ambiguity. Our intention in Proposition 1(c) was to define the interpolation inequality
>
> $\lVert A\rVert_p \le \lVert A\rVert_1^{1/p}\,\lVert A\rVert_{\infty}^{\,1-1/p}.$
>
> for $1 \le p \le \infty$, where we needed the conventions $1/\infty = 0$ and $t^{0}=1$
> for $t>0$ to include the endpoint cases $p=1$ and $p=\infty$. We agree that this extra notation is unnecessary and potentially confusing. In the revised manuscript, we have removed these
> conventions and now state the interpolation inequality explicitly for
> $1 < p < \infty$. This does not change any other result in the manuscript.
>
> **3. [Example 2, page 5]: this is labelled ``Example 2'' but is the first example to appear.**
>
> You are right. We have relabelled this as ``Example 1'' and updated all
> cross-references accordingly (also noted by Reviewer 5VLW).
>
> **4. [end of page 6]: the sentence ``Such refinements are vital, as they directly impact theoretical robustness guarantees for attention-based architectures.'' may be an overstatement.**
>
> We agree that the wording was too strong. In the revised version, we have changed this to
> 'Such refinements can directly impact theoretical robustness guarantees for attention-based architectures.'
>
> **5. [Fig.~2]: It would be nice to move Figure 2 into the Section on vision models, if possible.**
>
> Thank you for the suggestion. We have moved Figure 2 into the vision models section while keeping the layout compact.
>
> **6. [A.2 line 4 of text]: shouldn't $\boldsymbol{x}$ be in $\mathbb{R}^n$?**
>
> Yes. We have updated this.
>
> **7. [A.2 4th line of text]: should $\Delta^0_n$ be changed to $\Delta^\circ_n$ for consistency?**
>
> Yes, this was a typographical error. We have corrected this.
>
> **8. [Appendix]: could be nice to add something at the end of proofs to better separate them (e.g., between proofs of Thm.~1 parts (a) and (b)).**
>
> We appreciate this suggestion. In the revised Appendix, we have added this.
>
> **9. [proof of Proposition 2(b) line 1]: change 'for any $1 < p < \infty$' to 'for arbitrary $1 < p < \infty$'.**
>
>  Changed.
>
> **10. [Lemma 4(b) equation]: should '$j = \\{1, 2\\}$' instead be '$j \notin \\{1, 2\\}$'?**
>
> Yes, thank you for pointing out this typo. Corrected.
>
> **11. [end of page 15]: 'proof of Lemma' $\to$ 'proof of Lemma 4'.**
>
>  Changed
>
> **12. [3rd last text line on page 15]: $M(\hat{s})_{ij}$ should be equal to $-1/4$.**
>
> Yes, thank you for pointing out this error. Corrected.
>
> **13. [Proof of Lemma, case (ii), page 16]: change 'attained if' to 'attained only if'.**
>
>  Changed
>
> **14. [Last equation on page 16]:  redundancy: the exact same equation is repeated twice (I assume you meant to switch $i_3$ with $i_2$ in the second equation. Also, if I am not mistaken, you don't need to mention $i_3$ at all but can just say that $\boldsymbol{s}_{i_2} < 1/2$
> , right? Might be useful to do this for conciseness**
>
> Thank you for this careful observation. We agree that we need only $i_2$ to finish the proof and have updated the draft accordingly.
>
> **15. [Related work]: Mention the work by Kim, Papamakarios, and Mnih (2021) on the Lipschitz constant of self-attention and relate it to our result.**
>
> We appreciate this pointer. In the revised manuscript, we now cite this work and add a description explaining their result, which shows that the self-attention map is not globally Lipschitz on an unbounded domain, even though the softmax component inside the map is Lipschitz. Please refer to Section $4.2$ in the revised manuscript.
>
> **16. Broader impact: The current statement that there are ``no possible repercussions'' is too strong; mention both potential positive and negative impacts.**
>
> We agree that our original broader impact statement was too categorical. We have replaced it with a more nuanced discussion with the help of different works, including the works suggested by the reviewer. The current broader impact section emphasizes both positive and negative impacts.

---

> > ### Comment · Reviewer_GXEw · 2025-12-12
> > **response to authors' response**
> >
> > Thanks to the authors for the updates! I really appreciate their point-wise addressing of my concerns, particularly that they thoughtfully updated the broader impact section and that they added 4.2 to address point 15. Please note that I have not checked the new "Theorem 2" regarding the Cosine similarity attention--I am relying on the authors to double check this for any typos ;) ).
> >
> > In closing, thank you to the authors for submitting a solid paper--a paper that was a pleasure to review.

---

> > > ### Author Response · Authors · 2025-12-29
> > > **Second Response to  Reviewer GXEw**
> > >
> > > We thank the reviewer for the comments and the valuable clarifications.

---

### Review · Reviewer_5VLW · 2025-11-17

**Summary Of Contributions:**

This paper shows that the Lipschitz constant bound for Softmax with parameter $\lambda$, i.e., $\sigma_\lambda$ denoted by the authors, is exactly 1/2, under any $p$-norm induced operator norm. This is an improvement of e.g. Yudin et al.'s work which shows that in $\ell_2$ norm the constant is upper bounded by 1/2. Then the authors use the refined constant results to strengthen several existing results where Softmax continuity is relevant, covering e.g. zero-sum games and LLM.

**Additional Comments:**

N.A.

**Audience:**

Yes

**Audience Explanation:**

The paper strengthens several existing results related to machine learning.

**Broader Impact Concerns:**

N.A.

**Claims And Evidence:**

Yes

**Claims Explanation:**

The paper proves the results correctly. I have checked the Appendix.

**Requested Changes:**

The paper is mainly technical, so as long as it is technically sound, there is not much need to make changes. So I only have two questions:

1. In Appendix A.4, I believe the proof should be for Theorem 4 instead of Theorem 3. The title for Appendix A.4 may be incorrect.
2. In Theorem 3, I do not understand what the authors want to say by writing "$|W^K|_2 + n\nu \tau\varepsilon^{-1/2} |W^Q |_2 + ... $".

---

> ### Author Response · Authors · 2025-11-25
> **Response to Reviewer 5VLW**
>
> We thank the reviewer for carefully checking the Appendix, confirming the correctness of our proofs,
> and for the helpful comments regarding Appendix A.4 and the statement of Theorem 2.
> 1. **The title for Appendix A.4 may be incorrect.**
>
> Yes, Appendix A.4 contains the proof of our third Theorem, which was mistakenly labelled as
> Theorem 4 due to a numbering mix-up between theorems and examples (also noted by Reviewer
> GXEw). In the revised version, we have renumbered this result as Theorem 3 and updated the Appendix
> A.4 (title and cross-references) accordingly.
>
> 2. **In Theorem 3, I do not understand what the authors want to say by writing
> $\lVert W^K\rVert_2 + nντε^{−1/2}\lVert W^Q\rVert_2 + . . . .$**
>
> Thank you for pointing out that the statement of Theorem 3 (now Theorem 2 in the revised manuscript) was unclear. We intended to refine the explicit upper bound on the $\ell_2$-Lipschitz constant of the SCSA map (the Lipschitz-controlled variant of self-attention), given in the LipsFormer work by Qi et al. (2023):
>
> $L_2(SCSA) ≤ n^2 ν τ ε^{−1/2} \lVert W^K\rVert_2 + n ν τ ε^{−1/2} \lVert W^Q\rVert_2 + 2n ν ε^{−1/2} \lVert W^{V^⊤}\rVert_2.$
>
> In the original draft, this bound was written with an equality sign, which made the statement difficult to interpret. In the revised manuscript, we have rewritten Section 4.2 and now state the theorem using the inequality above, with $L_2(SCSA)$ explicitly defined as the global $\ell_2$-Lipschitz constant of the SCSA map.

---

### Decision · Action_Editor_bMqt · 2025-12-21

**Recommendation:** Accept as is

**Audience:**

Yes

**Audience Explanation:**

The paper is studying a topic relevant to majority of TMLR audience.

**Claims And Evidence:**

Yes

**Claims Explanation:**

The claims are supported by proofs and numerical results.